# Genome-Wide Association Analyses and Population Verification Highlight the Potential Genetic Basis of Horned Morphology during Polled Selection in Tibetan Sheep

**DOI:** 10.3390/ani14152152

**Published:** 2024-07-24

**Authors:** Dehong Tian, Zian Zhang, Bin Huang, Buying Han, Xue Li, Kai Zhao

**Affiliations:** 1Qinghai Provincial Key Laboratory of Animal Ecological Genomics, Key Laboratory of Adaptation and Evolution of Plateau Biota, Northwest Institute of Plateau Biology, Chinese Academy of Sciences, Xining 810001, China; 2University of Chinese Academy of Sciences, Beijing 100049, China; 3Qinghai Sheep Breeding and Promotion Service Center, Gangcha 812300, China

**Keywords:** GWAS, horn type, artificial selection, haplotype, population verification

## Abstract

**Simple Summary:**

The horned type of sheep, including the polled phenotype and scurs phenotype, emerged during the process of artificial domestication. Sheep serve as an optimal animal model for investigating the correlation between quality and quantity traits of horns, as well as the multi-gene regulatory mechanism underlying these traits. While significant progress has been made in understanding the effects of natural selection and sex selection on horn phenotypes, there is limited research on polymorphisms resulting from artificial selection. Through genome-wide association analysis, this study reveals the genetic mapping of horn phenotypes, polled genes, and superior haplotypes to provide valuable insights into studying the genetic mechanisms governing horn traits and other specific characteristics in ruminants while also laying a foundation for breeding.

**Abstract:**

The types and morphology of sheep horns have been extensively researched, yet the genetic foundation underlying the emergence of diverse horn characteristics during the breeding of polled Tibetan sheep has remained elusive. Genome-wide association analysis (GWAS) was performed on 103 subtypes (normal large horn, scurs, and polled) differentiated from G2 (offspring (G2) of parent (G1) of polled) of the polled core herd. Six single nucleotide polymorphisms (SNPs) located on chromosome 10 of the relaxin family peptide receptor 2 (RXFP2) gene exhibited positive correlations with horn length, horn base circumference, and horn base interval. Furthermore, in genotyping 382 G2 individuals, significant variations were observed for each specific horn type. Three additional mutations were identified near the target SNP upstream of the amplification product. Finally, the RXFP2-specific haplotype associated with the horned trait effectively maintained horn length, horn base circumference, and horn base interval in Tibetan sheep, as confirmed by population validation of nine loci in a sample size of 1125 individuals. The present study offers novel insights into the genetic differentiation of the horned type during improvement breeding and evolution, thereby establishing a robust theoretical foundation for polled Tibetan sheep breeding and providing valuable guidance for practical production.

## 1. Introduction

In cases where strong directional selection is evident, genetic traits often fail to exhibit the expected evolutionary response [1]. Therefore, elucidating the genes and genomic regions that contribute to trait variation provides an invaluable opportunity for comprehending the intricacies of genetic diversity processes within populations. The presence of adaptive variation at the genotype level may not be readily apparent when focusing on phenotypes for analysis, potentially due to genetic correlations between the phenotype of interest and other fitness-related traits [2]. Genetic resources obtained through directional selection [3] were utilized in this study to identify the genetic polymorphisms associated with horn traits within the population. Additionally, the effects of genetic variation on specific traits were determined, thereby establishing a crucial foundation for understanding the relationship between genotype and phenotype traits following artificial intervention.

The cranial appendages of ruminant animals, commonly referred to as headgear, consist of a pneumatized bony core fused with the frontal bone of the skull and are covered by a continuously growing keratin sheath [4]. Horn injuries to cattle caused by too-frequent and severe bumps are listed as one of the top 10 challenges to the U.S. beef supply [5]. To reduce injury risk to handlers and other animals as well as economic losses, dehorning procedures are performed [6]. Therefore, these appendages have become unpopular in modern captive breeding programs. The market-oriented sheep industry values its economic value because of animal welfare concerns [6,7,8]. More and more artificial interventions have been employed in recent years to genetically breed sheep without horns; the occurrence of this has been documented in historical records dating back to ancient Egypt [9]. However, there are still a few native breeds that naturally possess this trait. Prolonged human-mediated artificial selection has resulted in phenotypic variation within the genetic structure, which can be identified through patterns of genomic variation such as reduced nucleotide diversity and regions with very low heterozygosity. Importantly, the detection of numerous single nucleotide polymorphisms (SNPs) provides crucial information for identifying potential sites responsible for phenotypic variation. Therefore, the comprehensive identification and strategic deployment of specific SNPs via whole-genome sequencing (WGS) analysis promises substantial advancements in the domains of animal husbandry and molecular biology, particularly for the enhancement of production traits. A growing number of researchers are conducting extensive studies of various patterns of the presence and absence of horns, multi-horns, and deformed horns from around the world. With human domestication, domestic sheep breeds have also developed a hornless phenotype; breeders have been breeding polled sheep since the 16th century [10]. Earlier studies proposed that the autosomal 10 locus Ho^P^, Ho^+^, and Ho^hL^ alleles inheritance model controls the phenotype of horns, with the Ho^P^ allele determining polledness [10,11,12]. The horn allele has been mapped to OAR 10 such that there are at least two loci affecting the presence or absence of horns in Merino × Romney crosses [13]. The horn locus has been fine mapped to a 50 kb region in the sheep genome and predictive haplotypes for the polled HP allele have been identified using a segregating (Merino × Romney) × Merino resource [13]. RXFP2 is the principle candidate locus which supports a new model of horn-type inheritance in Soay sheep [14]. The analysis of the sequence revealed a 1833-base-pair genomic insertion located in the 3′-UTR region of the RXFP2 gene in polled animals [15,16]. The main candidate gene associated with the type and size of sheep horns, RXFP2, was found to have several new mutation sites. Among these mutations, the synonymous mutation p.P375 (c.1125A>G) was identified as a potential indicator for determining the presence or absence of horns in Tan sheep [17]. So far, the impact of two causal mutations on the process of horn formation remains unknown and, despite their influence on polled traits [18], no genetic variation has been identified that is causative of horn phenotype. Although the genetic loci associated with polledness have been mapped successfully via WGS, the observed variations in hornless sheep populations across different breeds are notable. Additionally, horn type in certain breeds is associated with variations in the RXFP2 gene region, indicating a highly complex genetic mechanism underlying the polled trait. These studies have only explained a small portion of the overall genetic variation; therefore, further experimentation is necessary to identify any causal sites related to the polled trait in ruminants.

Tibetan sheep are one of the prominent animal breeding resources unique to the Qinghai–Tibet Plateau. In order to elucidate the genetic characteristics of horn development, we established polled male and female groups (G1) based on phenotype in the adult phase and performed targeted mating, with the resulting offspring (G2) serving as our research subjects. Until now, there have been no reports on the genetic basis underlying horned differentiation enhanced breeding.

In this study, we conducted GWAS to analyze three horn phenotypes in G2 and subsequently identified specific haplotypes through population verification. By investigating candidate genes associated with polled traits at the genomic level, we were able to assess the molecular genetic characteristics of polled traits and provide fundamental data for localizing functional genes related to horn traits. This will serve as a crucial reference for the future molecular breeding of polled sheep.

## 2. Materials and Methods

### 2.1. Samples Collection and Whole-Genome Sequencing

The core group exhibited phenotypic variations in the second generation (G2), where polled rams and ewes (G1) were mated resulting in individuals with either normal large horns, scurs, or no horns. After continuously observing the horn status of a core population with a well-defined pedigree, we meticulously selected a total of 103 sheep for sequencing analysis, including 48 polled sheep, 29 sheep with normal large horns, and 26 sheep with scurs. The genotyping of 382 G2 individuals was conducted within a non-horned core group. The phenotypic values of the compiled horns were collected, and their whole blood was obtained and stored in EDTA vacuum tubes. The husbandry and management of these animals was consistently maintained at the same level. The QIAamp DNA Mini Kit (Qiagen, Hilden, Germany) was utilized to extract total genomic DNA from 200 µL of sheep blood, followed by assessment of DNA quality. Illumina’s paired-end DNA Sample Prep kit standard library building procedure was employed to prepare sequencing libraries using a minimum of 3 µg genomic DNA. The Agilent 2100 bioanalyzer (Agilent, Santa Clara, CA, USA) was used for examination and real-time PCR quantification of the sequencing libraries prior to sequencing on the Illumina NovaSeq6000 NGS platform (Guangzhou, China).

### 2.2. Quality Control and Annotation

The Illumina data underwent filtration using FASTP (version 0.18.0) with the following criteria: (1) reads containing ≥10% unknown nucleotides (N) were removed; (2) reads with ≥50% of bases having Phred quality scores ≤ 20 were removed; and (3) reads with adapters were deleted. The resulting filtered clean reads were utilized for assembly analysis. The filtered reads were aligned to the reference genome using the Burrows–Wheeler Aligner (BWA) (version 0.7.12) alignment software (ncbi_GCF_002742125.1) with the settings ‘mem 4 - k 32 - M’, - k as the minimum seed length, and -M as an option used to mark shorter split alignment hits as secondary alignments [19]. Variant calling was performed for all samples using GATK’s Unified Genotyper (version 3.4-46) [20]. SNPs and InDels were filtered using GATK’s Variant Filtration with proper standards (-Window 4, -filter QD < 2.0 || FS > 60.0 || MQ < 40.0, -G_filter “GQ < 20”), and those exhibiting segregation distortion or sequencing errors were discarded. To determine the physical positions of each SNP, the variants were annotated using the ANNOVAR software (version 2) tool [21] to align and annotate SNPs.

### 2.3. Genome-Wide Association Analysis

The implementation of REGENIE [22] and SAIGE [23] was chosen due to the presence of rare variants and highly imbalanced case–control ratios. In an effort to enhance the power of the association test, we employed a gene-region-based approach (or other methods for generating variant sets) and utilized GMMAT (version 0.9.3) [24] software for conducting gene/set-based association analysis. We included three tests in the framework: the burden test, SKAT (sequence kernel association test) [25], and SKAT-O (optimal sequence kernel association test) [26]. The SKAT method employed a multiple regression model to directly assess the relationship between the phenotypes and genetic variants in a variant set, as well as covariates. This approach allows for variations in both direction and magnitude of effects among different variants. In the burden test, SKATO estimated the weight and SKAT statistics that maximize power using the formula T_SKAT−0_ = (1 − ρ)T_SKAT_ + ρT_burden_. Additionally, SKAT-O required a grid search over ρ to determine the minimum *p*-value. The Bonferroni correction threshold (*p*-value = 0.01/marker number or 0.05/marker number) was employed to identify statistically significant associations. Candidate genes (CAGs) located within a 50 kb region upstream or downstream of the significantly associated markers were identified.

### 2.4. Enrichment Analysis of Candidate Genes

The candidate genes were mapped to GO terms in the Gene Ontology database [27] (http://www.geneontology.org/, accessed on 20 April 2024). Gene numbers were calculated for each term, and significantly enriched GO terms were determined using a hypergeometric test. The *p*-value calculation formula is as follows:P=1−∑i=0m−1MiN−Mn−iNn

The number of all genes with GO annotation is represented by N, while n represents the number of genes/CAGs in N. Similarly, M denotes the number of all genes annotated to specific pathways, and m represents the number of genes/CAGs in M. The calculated *p*-value underwent false discovery rate (FDR) correction, with a threshold set at FDR ≤ 0.05. Pathways that meet this condition are defined as significantly enriched GO terms in genes/CAGs.

### 2.5. Group Validation of Significantly Correlated SNPs

The blood samples of 1125 sheep from a large group were collected in a random manner, and the horn phenotype data (Figure 1, Appendix A), including measurements of horn length, horn base circumference, and horn base interval, were recorded using the following methods:

Horn length: the arc length along the longitudinal edge of the back of the horn, from its base to its end;

Horn base circumference: the circumference of the longitudinal base encircling the horn base;

Horn base interval: the linear distance between the vertical bases on the posterior side of the horn base.

The nine SNPs detected in exons of genes identified by screening analysis were validated through Sanger sequencing and SNaPshot mini-sequencing, and primers were designed using Primer3 v0.4.0 (1) [28] while single-base extension primers were designed using the online tool Primer 3 plus (Appendix A).

The amplification and screening of SNPs were conducted in 1125 sheep to analyze the correlation between candidate mutations and horn traits. Initially, the amplification of exons from genomic DNA was achieved using primers. PCR reactions were performed in 10 μL volumes containing 5 μL Taq PCR Mix, 1 μL Primer Mix, 1 μL DNA, and 3 μL ddH_2_O. A PCR program consisting of an initial denaturation at 94 °C for 5 min, followed by 35 cycles at 94 °C for 20 s, 60 °C for 30 s, and finally at 72 °C for 30 s was carried out using a Veriti™96-Well Thermal Cycler (Applied Bio-systems, Foster City, CA, USA). All PCR products were purified with ExoSAPIT^®^ (USB Corporation, Cleveland, OH, USA) at 37 °C for 40 min followed by incubation at 85 °C for 15 min. Subsequently, all SNaPshot^®^ reactions were performed in a volume of 5 μL including 0.5 μL SNaPshotMix, 3 μL Pooled PCR Products, 1 μL Pooled Primers, and 0.5 μL ddH_2_O with a program consisting of an initial denaturation at 95 °C for 30 s and 35 cycles comprising 95 °C for 5 s, 52 °C for 5 s, and 60 °C for 5 s. All SNaPshot^®^-specific products were purified with Shrimp Alkaline Phosphatase-SAP (Thermo Fisher Scientific Inc., Waltham, MA USA) and incubated at 95 °C for 5 min. Finally, the Genemarker Software (v. 1.90) was used to analyze all the SNaPshot^®^-specific results. Allele frequency and genotype frequency were assessed using web-based software to determine their adherence to the Hardy–Weinberg equilibrium (http://scienceprimer.com/hardy-weinberg-equilibrium-calculator, accessed on 20 April 2024).

The association between various genotypes or haplotypes and horn traits was assessed using a sampling approach based on general linear models:Y_ijTl_ = ξ + α_i_ + β_j_ + γ_T_ + δ_l_ + θ

Here:

Y_ijTl_ is the horned trait measured observation;

ξ is the overall mean;

α_i_ is the genetic effect;

β_j_ is the individual age effect;

γ_T_ is the year effect;

δ_l_ is the permanent environment effect;

θ is the random residual effect and is assumed to be independent, N (0, σ2) distribution.

The data were analyzed using the General Linear Model (GLM) in SPSS Statistics (version 29, IBM, Armonk, NY, USA), and statistical significance was determined at a *p*-value of less than 0.05.

## 3. Results

### 3.1. Summary Statistics of Phenotype Data and Quality Control

We established a polled core population and observed polymorphisms in the horn phenotypes of G2 generation sheep through directional selection, including degenerated, deformed horns known as scurs, polled individuals, and individuals with normal large horns. The descriptive statistics of phenotypic values associated with horn-type traits in a population of 103 Tibetan sheep are provided in Appendix A.

After implementing quality control and management measures for 36,023,978 SNP loci, the following conditions were excluded: (1) 14,513,444 SNPs with a secondary gene frequency less than 0.05; (2) 200,718 SNPs with a deletion rate exceeding 0.5; and (3) 30,361 SNPs with a heterozygosity ratio greater than 0.8. Subsequently, a total of 21,279,455 independent SNPs remained for the subsequent analysis. The final count revealed a total of 33,065,497 single nucleotide polymorphisms (SNPs) and 2,958,481 insertions/deletions (indels). The majority of the high-quality single nucleotide polymorphisms (SNPs) (61.61%) were observed in intergenic regions, characterized by T/C and A/G substitutions, while only 0.70% were located within exonic regions. The remaining SNPs were situated upstream (0.61%) and downstream (0.63%) of the open reading frame, primarily found within intronic regions (33.97%) (Appendix A).

### 3.2. Population Structures and Genome-Wide Association Analysis

According to the population structure, admixture v1.3 was employed with a range of K values from 2 to 5 (Figure 2A). Specifically, a K value of 2 was utilized and the corresponding outcome is depicted in Figure 2B. Kinship estimation and principal component analysis (PCA) were conducted on all individuals, confirming the validity of the sampling (Figure 2C,D). The bivariables in the Kinship plot are perfectly aligned along a straight line, indicating a strong linear relationship. Moreover, the sample exhibits high repeatability, suggesting excellent data reproducibility.

According to the number of independent effective SNPs, the genome-wide significance and suggestive significance values were determined as a *p*-value of 4.06 × 10^−10^ corresponding to a 1% significance level and a *p*-value of 2.34 × 10^−9^ corresponding to a 5% significance level, respectively. SNPs with *p*-values lower than these thresholds were identified as being associated with Tibetan sheep horn traits. The *p*-value Manhattan map shows significant correlations between SNPs and horn length (Figure 3A), horn base circumference (Figure 3C), and horn base interval (Figure 3E), with statistically significant regions covering SNPs sites (Figure 3B,D,F). These aforementioned SNPs exhibit a strong positive correlation with the three horn-related traits (Table 1, Appendix A), surpassing the predetermined threshold. Further verification will be conducted subsequently. Concurrently, we conducted a comparison of the reported sites in the literature pertinent to RXFP2 and observed that only a single site (OAR10_29461968) aligned with our findings (Table 2).

The genetic polymorphism of horn type was observed in both sexes, with three distinct morphological characteristics: large normal horns, scurs, and polled (polled accounted for 31.12% and 54.55% of all recorded cases in male and female sheep, respectively) (Table 3). Each horn type exhibited a significant number of quantitative variation sites. Differences in the dominance and expression of specific alleles were evident between the sexes, posing challenges to inferring an individual’s genotype.

### 3.3. Enrichment Analysis

To further elucidate candidate genes associated with significant SNPs, we conducted additional gene enrichment analysis. The pathways enriched in the RXFP2 gene were further subjected to additional recordings. The GO/KEGG pathway analysis revealed the presence of certain pathways associated with horn characteristics. Specifically, the relaxin signaling pathway and neuroactive ligand–receptor interaction were found to be significantly KEGG co-enriched in terms of horn length, horn base circumference, and horn base interval. Furthermore, these three traits collectively exhibited enrichment in 33 GO term pathways. The candidate genes associated with horn length (Figure 4A, Appendix A), horn base circumference (Figure 4B, Appendix A), and horn base interval (Figure 4C, Appendix A) exhibit co-enrichment during gonad development, oocyte maturation, sex differentiation, protein–hormone receptor activity, cell maturation and differentiation processes, as well as positive regulation of CAMP-mediated signaling pathways. These findings suggest that these three traits are likely regulated by shared biological pathways. The enrichment analysis revealed that the RXFP2 gene exhibited the highest level of enrichment among the Gene Ontology terms associated with biological processes.

### 3.4. Population Verification of SNPs Significantly Correlated with Tibetan Sheep Horn Traits

The SNPs identified by the GWAS mentioned above were subjected to significant testing between horned and polled Tibetan sheep, revealing a significant correlation between the RXFP2 gene and polled Tibetan sheep, which has been extensively reported in the literature. Furthermore, the descriptive statistical information of phenotypic values obtained from a large population through the random selection of 1125 ewes (Appendix A) provided population verification for these polymorphisms at the aforementioned RXFP2 sites. Genotyping, population genetic analysis, and association determination between SNPs and horn traits were conducted for all loci.

GWAS predicted six individually detected SNPs, while three additional mutations were identified in close proximity to the target SNPs upstream of amplification products (Appendix A). The value of PIC at SNP001 was <0.25, which was indicative of low polymorphism. The other SNPs we identified exhibited moderate levels of polymorphism (0.25 < PIC < 0.5) (Appendix A).

Nine SNP verification results revealed significant correlations between three mutation sites and the horn length, horn base circumference, and horn base interval of Tibetan sheep (Table 4). The genotype of horns had a significant effect on horn length, horn base circumference, and horn base interval, while there might be a polygenic influence associated with the genotypic horn.

### 3.5. Linkage Disequilibrium and Haplotype Block Association Analysis

The analysis results of linkage disequilibrium (LD) indicate a strong correlation between the SNP sites of the two haplotype modules (D’ > 0.85), with a coefficient of 0.81 observed for the correlation between the two haplotype modules (Figure 5A). Subsequently, a correlation analysis was conducted to examine the haplotypes of the two modules, revealing significant effects of different haplotypes on the horn type of Tibetan sheep (Table 5 and Table 6). In the 19 kb LD block, twelve haplotypes containing three SNPs were identified, while in the 51 kb LD block, twenty-two haplotypes containing six SNPs were found (Figure 5B,C). In Module 1, three haplotypes, namely AAGGCT (*n* = 11, *p*-value < 0.01), GAGACT (*n* = 9, *p*-value < 0.01), and GAGGTT (*n* = 8, *p*-value < 0.01) (Table 5), exhibited significant correlations with polled traits. In Module 2, the haplotypes AACCAAGGCTCT (*n* = 2, *p*-value < 0.01), AATTGAAATTTT (*n* = 18, *p*-value < 0.01), AATTGGAATTTT (*n* = 3, *p*-value < 0.01) GACTGAGATTCT (*n* = 4, *p*-value < 0.01), and GATTGGAATTTT (*n* = 15, *p*-value < 0.01) (Table 6) were found to be significantly associated with polled traits. The maintenance mechanism of horn characteristic variation is determined by the co-breeding of haplotypes associated with these three traits.

## 4. Discussion

The vestigialization of horns occurs in most domestic lines due to the diminishing relevance of traits that ensure fitness in natural environments under artificial breeding conditions [30]. Domestic animals have undergone domestication and improvement to produce similar phenotypic transformations [34], a phenomenon known as “domestication syndrome” [35,36,37]. However, our findings indicate that artificial intervention has led to the differentiation of three distinct horn phenotypes (normal large horns, scur, and polled) in polled-orientated progeny. Furthermore, we have determined that all types of horns, including horn length, horn base circumference, and horn base interval, are regulated by the RXFP2 gene located on chromosome 10 [33,38]. We can distinguish phenotypes that have the same horn but have different genotypes, suggesting that the genotype of the horn may be responsible not only for discrete variation in the diagonal type, but also for many quantitative genetic variations in the diagonal size [39], and that the quantified variation differences in the size of the horn appear to be caused by the genotype of the horn. This study establishes a foundation for investigating genotypic-level selection of horn types and morphology and enhances our understanding of the mechanisms underlying the maintenance and differentiation of horn polymorphisms in this population.

Interestingly, horn polymorphisms also occur in natural populations, such as Soay sheep (males exhibit both normal and scurred horns, while females can have normal, scurred, or polled horns) [39]. The same phenomenon is observed in the natural population of Tibetan sheep. It remains unclear whether this phenotype is associated with genetic introgressions or a free combination of RXFP2 superior alleles linked to horn types within the population. This hypothesis can be tested in future studies by generating genotypic data for additional sheep breeds.

The development of horns in sheep is influenced not only by genetic and sexual factors, but also by hormonal and environmental influences. Seasonal horn growth typically occurs during spring, when prolactin secretion reaches its peak. There are notable variations in the timing and pace of horn development across different life stages [40]. Sustained low levels of testosterone during the inactive period are essential for promoting male-type horn growth, while high levels of testosterone prior to estrus can actually impede growth—a mechanism that also regulates antler development in deer [41,42]. Sheep exhibited cryptorchidism after the RXFP2 gene was knocked out using CRISPR/Cas9 technology [43]. The growth rate of sheep horns is regulated by thyroid activity, with the peak concentration of thyroid hormone (T3, T4) coinciding with the rapid growth period [44]. Our findings demonstrated a significant correlation between the concentration of RXFP2 and gonadal development and maturation, as well as the regulatory impact of reproductive hormones on sheep horn development. Further investigation into the hormonal regulatory mechanisms will be conducted.

The inheritance of horns in sheep is closely associated with sex, as indicated by Wood’s findings [45]. Specifically, the horned and hornless genes exhibit distinct effects on males and females. In this regard, it can be observed that large horns are dominant in males while recessive in females [45]. A study conducted by Dominik et al. revealed the presence of a single nucleotide polymorphism located in a specific genomic region, which exhibited a prominent maternal imprinting effect in both males and females [12]. QTL analysis of diagonal morphology revealed a significant interaction between horn shape traits and sex, indicating that the same chromosomal region accounted for genetic variation in both horn length and base circumference angle. Furthermore, it was determined that these two traits were subject to genotypic (synergistic) selection [38]. The fixation of the ideal individual’s characteristics can be rapidly achieved by mating a homozygous hornless male with a homozygous hornless female [45], resulting in all offspring being hornless.

Prediction within a single family will incorporate both linkage information and LD information, assuming that non-genetic factors have been accurately accounted for. However, in the case of cross-family prediction, reliance can only be placed on LD information [46]. Haplotypes encompass a greater amount of information (heterozygosity) compared to a single SNP, thereby significantly enhancing accuracy and robustness in gene localization. Conversely, in small sample sizes, a single SNP fails to establish an association between a specific phenotype and gene, whereas haplotypes possess this capability [47]. Horn phenotypes can be accurately predicted with an accuracy of approximately 0.7 when utilizing a SNP for prediction [33]. Consequently, investigations into phenotypic correlation with genetic variation are more likely to yield successful outcomes if the SNPs employed have been demonstrated to exhibit linkage imbalance through methodologies such as haplotype analysis [48,49,50]. The alleles located on OAR10 at position 29,461,968 were found to be genetically linked with the neighboring loci in the gene map [14,30]. LD was observed to be lower between the 1.78 kb insert fragment of the RXFP2 gene and the SNP located at OAR10_29511510.1 (position 29,476,678) [16], whereas a higher LD was observed between the SNP inserted with OAR10_29546872.1) [33]. In our study, the haplotype combinations of AAGGCT, GAGACT, and GAGGTT as well as AACCAAGGCTCT, AATTGAAATTTT, AATTGGAATTTT, GACTGAGATTCT, and GATTGGAATTTT are the predominant ones. The presence of fitness disparities may also occur among sheep exhibiting similar phenotypes but possessing different underlying genotypes due to their association with other alleles at adjacent sites in a gene-linkage imbalance. We have successfully identified the haplotype associated with the horn trait, indicating that the absence of horns in artificially bred sheep is influenced by the interaction of multiple SNPs. This finding also highlights the synergistic genetic basis underlying these three horn traits. It is important to note that single SNP loci alone cannot provide sufficient evidence for establishing a correlation between variations in the RXFP2 gene and the expression of the horn trait.

Polled rams exhibit relatively low breeding values and were predominantly heterozygous, the proportion of polled individuals has greater potential in the G2 population, where ewes outnumber rams. It is evident that there exists a common genetic basis for both horn type and length, with at least 76% of the genetic variation attributed to the “horn” genotype [14]. Consequently, genome selection can significantly accelerate the rate of genetic improvement and positively impact both the breeding value and quantity of polled rams. Quantitative trait genes in animals were directly selected using molecular marker technology, and whole-genome association analysis based on the genome variation map was employed to enhance breed improvement more effectively. A breeding program thus may well produce polled sire with breeding values comparable to the top sire in the near future [51]. Based on these findings, we anticipate that a majority of lambs born in generations G3–G4 will be polled, thus establishing a new prototype for a polled strain.

## 5. Conclusions

In summary, our GWAS identified six single nucleotide polymorphisms (SNPs) on the RXFP2 gene located on chromosome 10 that exhibited significant associations with horned traits. Each horn type displayed a substantial number of variation sites. Furthermore, three additional mutations were subsequently detected near the target SNP upstream of the amplification product. Population validation of the specific haplotype is considered indicative of distinguishing between horn and polled phenotypes while maintaining horn length, horn base circumference, and horn base interval. This study aims to provide novel insights into the molecular characteristics of polled sheep under human intervention and can offer practical guidance.

## Figures and Tables

**Figure 1 animals-14-02152-f001:**
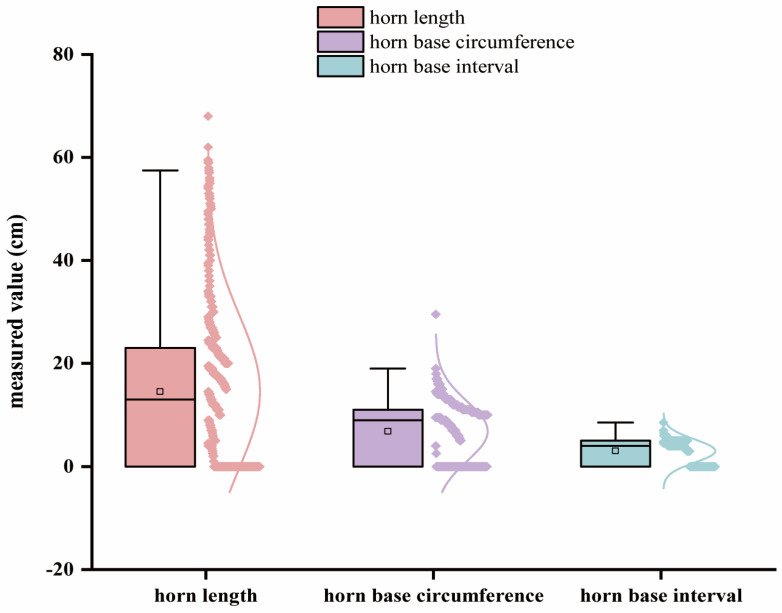
Measurement of horn length, horn base circumference, and horn base interval. The X axis represents the measured position, while the Y axis represents the measured value, which is depicted through a combination of a box plot and normal curve.

**Figure 2 animals-14-02152-f002:**
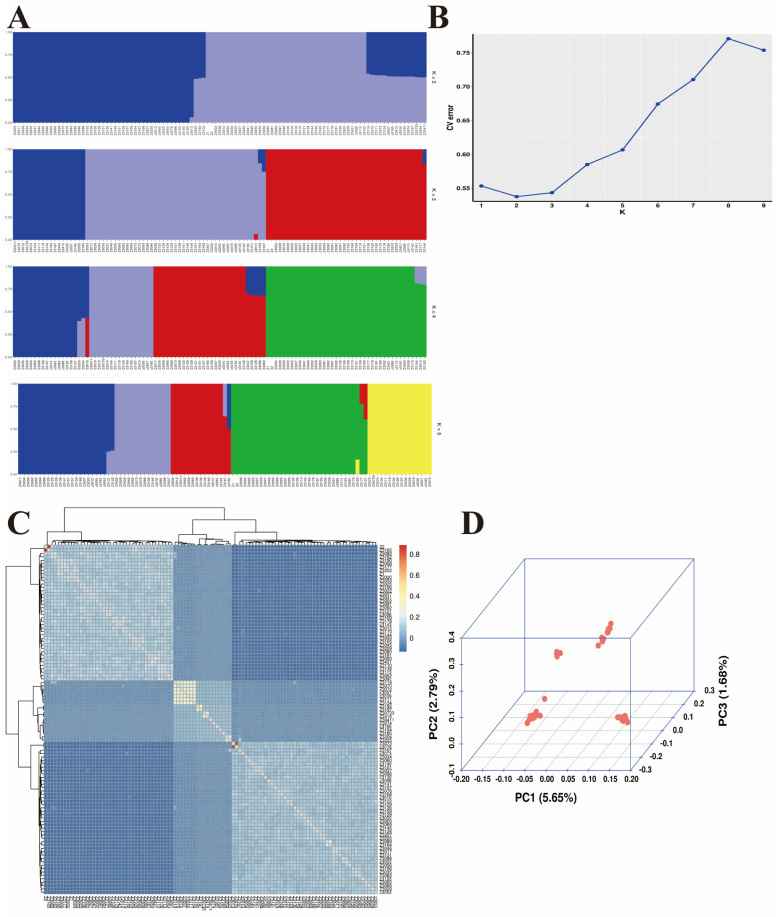
Population structure analysis in the nucleus herd of polled Tibetan sheep. (**A**) Population structure with K from 2 to 5; (**B**) cross-validation plot for determining the best K; (**C**) kinship plot of 103 Tibetan sheep; (**D**) principal component analysis.

**Figure 3 animals-14-02152-f003:**
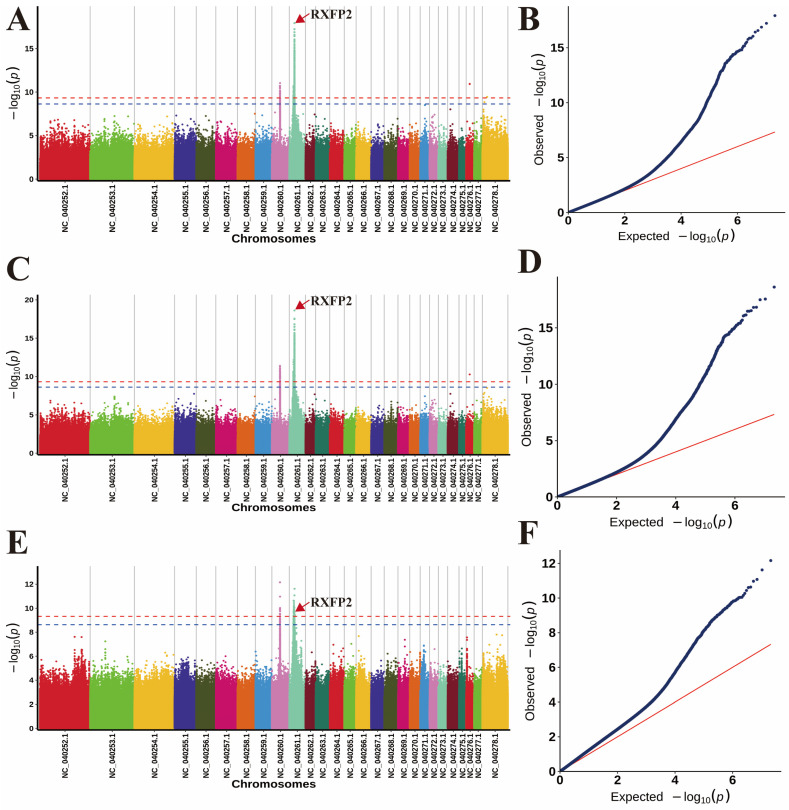
Manhattan plots and QQ plots of (**A**) horn length, (**C**) horn base circumference, and (**E**) horn base interval traits. (**B**) QQ plots of the horn length, (**D**) QQ plots of the horn base circumference, (**F**) QQ plots of the horn base interval.

**Figure 4 animals-14-02152-f004:**
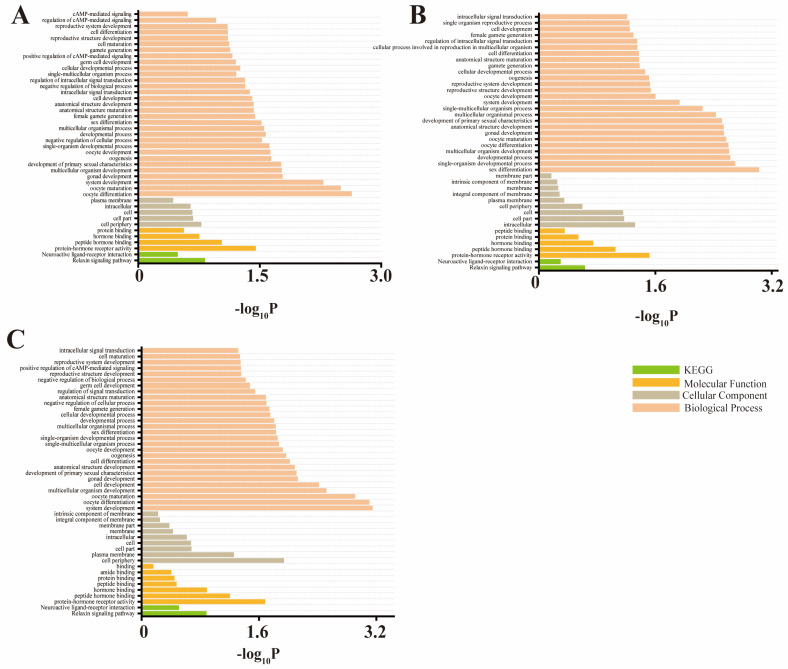
Gene Ontology (GO) and KEGG pathway analysis of the RXFP2 gene in terms of (**A**) horn length, (**B**) horn base circumference, and (**C**) horn base interval.

**Figure 5 animals-14-02152-f005:**
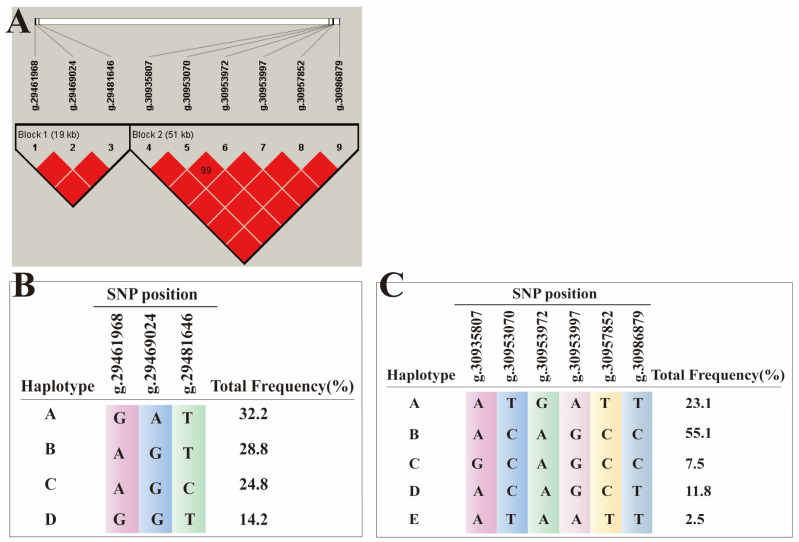
Linkage disequilibrium plot (r^2^) and haplotype blocks for SNPs of the RXFP2 gene. (**A**) A 19-kb LD block and 51-kb LD block; (**B**) four haplotypes within the sampled sheep population; (**C**) five haplotypes within the sampled sheep population.

**Table 1 animals-14-02152-t001:** Analysis of related SNPs associated with horn traits.

Trait	Chr	Related SNPs	Position (bp)	*p*-Value	Candidate Gene
horn length	10	NC_040261.1	30,953,972	1.19 × 10^−18^	RXFP2 within
			30,953,997	5.98 × 10^−18^	
			30,957,852	3.76 × 10^−17^	
			30,986,879	8.96 × 10^−17^	
			30,953,070	1.72 × 10^−16^	
horn base circumference	10	NC_040261.1	30,953,972	2.42 × 10^−19^	
			30,953,997	3.24 × 10^−18^	
			30,953,070	3.36 × 10^−17^	
			30,986,879	8.84 × 10^−17^	
horn base interval	10	NC_040261.1	30,935,807	6.24 × 10^−10^	

**Table 2 animals-14-02152-t002:** Analysis of RXFP2 site characteristics.

Traits	Loic	Current Name (Breed)	References
horn type	c.789C>T	Tan sheep	[17]
	c.1117A>G		
	c.1125A>G		
	c.2059C>T		
horn size	OAR10_29685536	Thinhorn sheep	[29]
horn shape	OAR10 29461968	89 Chinese sheep	[30,31]
horn size	29.45–29.55 M	Bighorn sheep	[32]
horn phenotype	OAR10_29415140	Soay sheep	[14]
	OAR10_29455959		
	OAR10_29511510		
polled phenotype	OAR10_29389966	Merino sheep	[12]
horn phenotype	OAR10_29448537	Merino sheep	[33]
	OAR10_29458450		
	OAR10_29546872		
horn phenotype	OAR10_29481646	Tibetan sheep	a, [31]
	OAR10_29469024		
	OAR10_30935807	Tibetan sheep	a
	OAR10_30953070		
	OAR10_30953972		
	OAR10_30953997		
	OAR10_30957852		
	OAR10_30986879		

a—this work.

**Table 3 animals-14-02152-t003:** Phenotypic distribution and potential genotypes of horn type in genetically matched polled offspring.

Sex	Horn Type	SNP	Genotype			Polled Frequency (%)
			Wild	Mutant	Homozygous	
Male (*n* = 193)	Normal	g.30935807	85	1	0	43.88
		g.30953070	1	23	62	
		g.30953972	0	24	62	
		g.30953997	1	23	62	
		g.30957852	1	25	60	
		g.30986879	4	43	39	
	Scurred	g.30935807	36	12	1	25.00
		g.30953070	3	34	12	
		g.30953972	1	36	12	
		g.30953997	3	34	12	
		g.30957852	3	34	12	
		g.30986879	13	27	9	
	Polled	g.30935807	41	19	1	31.12
		g.30953070	21	40	0	
		g.30953972	0	61	0	
		g.30953997	21	40	0	
		g.30957852	22	39	0	
		g.30986879	27	34	0	
Female (*n* = 189)	Normal	g.30935807	57	2	0	31.55
		g.30953070	0	13	46	
		g.30953972	46	13	0	
		g.30953997	0	13	46	
		g.30957852	0	15	44	
		g.30986879	1	28	30	
	Scurred	g.30935807	23	2	1	13.90
		g.30953070	2	8	16	
		g.30953972	0	10	16	
		g.30953997	2	8	16	
		g.30957852	2	8	16	
		g.30986879	3	15	8	
	Polled	g.30935807	77	25	0	54.55
		g.30953070	17	74	11	
		g.30953972	0	91	11	
		g.30953997	17	74	11	
		g.30957852	18	74	10	
		g.30986879	39	59	4	

**Table 4 animals-14-02152-t004:** Association analyses of SNPs and genotypes in RXFP2 with horn traits.

SNP	Genotype	Horn Length (cm)	Horn Base Circumference (cm)	Horn Base Interval (cm)
g.30935807	AA (962)	16.51 ± 14.77 ^Aa^	7.72 ± 5.57 ^Aa^	3.36 ± 2.09 ^Aa^
	GA (157)	2.85 ± 7.13 ^Bb^	1.61 ± 3.92 ^Bb^	0.93 ± 1.90 ^Bb^
	GG (6)	3.50 ± 4.28 ^ABb^	3.00 ± 4.69 ^ABb^	1.50 ± 2.35 ^ABb^
		*p* = 5.945 × 10^−21^	*p* = 2.3087 × 10^−17^	*p* = 0.008457
g.30953070	TT (72)	1.98 ± 5.57 ^C^	1.10 ± 3.29 ^C^	0.97 ± 1.77 ^C^
	CT (419)	8.65 ± 13.34 ^B^	4.14 ± 5.65 ^B^	2.13 ± 2.35 ^B^
	CC (634)	19.85 ± 13.95 ^A^	9.28 ± 4.73 ^A^	3.83 ± 1.79 ^A^
		*p* = 2.3448 × 10^−9^	*p* = 2.0722 × 10^−23^	*p* = 1.0313 × 10^−45^
g.30953972	GG (28)	2.88 ± 7.51 ^c^	1.55 ± 3.93 ^c^	0.57 ± 1.44 ^C^
	AG (465)	8.12 ± 13.10 ^Bb^	3.86 ± 5.56 ^Bb^	2.06 ± 2.32 ^B^
	AA (632)	19.78 ± 13.91 ^Aa^	9.26 ± 4.73 ^Aa^	3.83 ± 1.79 ^A^
		*p* = 0.012412	*p* = 6.8297 × 10^−15^	*p* = 7.1798 × 10^−49^
g.30953997	AA (73)	2.25 ± 5.99 ^C^	1.22 ± 3.43 ^C^	1.01 ± 1.80 ^C^
	GA (422)	8.86 ± 13.54 ^B^	4.21 ± 5.68 ^B^	2.15 ± 2.35 ^B^
	GG (630)	19.76 ± 13.92 ^A^	9.26 ± 4.73 ^A^	3.82 ± 1.80 ^A^
		*p* = 1.0693 × 10^−8^	*p* = 4.3225 × 10^−23^	*p* = 1.0569 × 10^−45^
g.30957852	TT (77)	1.85 ± 5.41 ^C^	1.03 ± 3.19 ^C^	0.96 ± 1.76 ^C^
	CT (424)	8.81 ± 13.30 ^B^	4.26 ± 5.67 ^B^	2.17 ± 2.35 ^B^
	CC (624)	20.00 ± 13.97 ^A^	9.31 ± 4.71 ^A^	3.84 ± 1.79 ^A^
		*p* = 1.7251 × 10^−10^	*p* = 2.2845 × 10^−26^	*p* = 1.267 × 10^−47^
g.30986879	TT (176)	3.33 ± 8.47 ^C^	1.77 ± 4.15 ^C^	1.08 ± 1.99 ^C^
	CT (490)	11.63 ± 13.25 ^B^	5.79 ± 5.73 ^B^	2.80 ± 2.32 ^B^
	CC (459)	21.94 ± 14.31 ^A^	9.90 ± 4.44 ^A^	3.98 ± 1.60 ^A^
		*p* = 5.5701 × 10^−14^	*p* = 1.2202 × 10^−39^	*p* = 5.9519 × 10^−57^
g.29469024	AA (131)	2.90 ± 8.96 ^C^	1.36 ± 3.78 ^C^	0.60 ± 1.57 ^C^
	GA (463)	12.31 ± 14.61 ^B^	5.80 ± 5.82 ^B^	2.54 ± 2.34 ^B^
	GG (531)	19.36 ± 13.88 ^A^	9.10 ± 4.92 ^A^	4.02 ± 1.560 ^A^
		*p* = 9.9762 × 10^−15^	*p* = 51.2842 × 10^−35^	*p* = 1.1715 × 10^−73^
g.29481646	TT (833)	16.56 ± 14.47 ^A^	7.82 ± 5.47 ^A^	3.45 ± 2.04 ^A^
	CT (27)	5.85 ± 11.51 ^B^	2.65 ± 5.12 ^B^	1.13 ± 1.98 ^B^
	CC (265)	9.05 ± 14.10 ^B^	4.20 ± 5.77 ^B^	1.85 ± 2.31 ^B^
		*p* = 1.4899 × 10^−14^	*p* = 0.01	*p* = 5.599 × 10^−11^
g.29461968	GG (306)	2.72 ± 8.85 ^C^	1.25 ± 3.64 ^C^	0.56 ± 1.53 ^C^
	AG (434)	16.70 ± 14.65 ^B^	7.88 ± 5.54 ^B^	3.45 ± 2.07 ^B^
	AA (385)	21.49 ± 12.84 ^A^	10.11 ± 3.97 ^A^	4.47 ± 0.87 ^A^
		*p* = 6.7264 × 10^−34^	*p* = 2.828 × 10^−86^	*p* = 75.0538 × 10^−47^

Within the same line, the mean values of different superscript lowercase letters were significantly different (*p* < 0.05). Within the same line, the mean values of different superscript uppercase letters differ significantly (*p* < 0.01).

**Table 5 animals-14-02152-t005:** Association analysis of haplotypes in RXFP2 with horn traits in Module 1.

Loci	Haplotypes	Numbers	Horn Length (cm)	Horn Base Circumference (cm)	Horn Base Interval (cm)
g.29481646 g.29469024 g.29461968	AAGGCT	11	0.00 ± 0.00 ^B^	0.00 ± 0.00 ^B^	0.00 ± 0.00 ^B^
	AAGGTT	120	3.16 ± 9.32 ^B^	1.48 ± 3.92 ^B^	0.65 ± 1.63 ^B^
	GAGACT	9	0.00 ± 0.00 ^B^	0.00 ± 0.00 ^B^	0.00 ± 0.00 ^B^
	GAGATT	324	16.46 ± 14.55 ^A^	7.79 ± 5.46 ^A^	3.41 ± 2.08 ^Aa^
	GAGGCC	122	2.99 ± 9.73 ^B^	1.34 ± 3.77 ^B^	0.59 ± 1.60 ^B^
	GAGGTT	8	0.00 ± 0.00 ^B^	0.00 ± 0.00 ^B^	0.00 ± 0.00 ^B^
	GGAACC	3	23.83 ± 20.52 ^A^	10.33 ± 4.75 ^A^	4.50 ± 1.00 ^Aa^
	GGAACT	7	22.57 ± 11.58 ^A^	10.21 ± 4.84 ^A^	4.36 ± 0.75 ^Aa^
	GGAATT	375	21.45 ± 12.83 ^A^	10.11 ± 3.96 ^A^	4.48 ± 0.88 ^Aa^
	GGGACC	96	19.53 ± 14.65 ^A^	9.12 ± 5.32 ^A^	4.00 ± 1.72 ^Aa^
	GGGATT	5	8.30 ± 11.85 ^A^	4.40 ± 6.12 ^A^	1.80 ± 2.49 ^Ab^
	GGGGCC	44	2.00 ± 6.68 ^B^	0.98 ± 3.20 ^B^	0.48 ± 1.38 ^B^

Within the same line, the mean values of different superscript lowercase letters were significantly different (*p* < 0.05). Within the same line, the mean values of different superscript uppercase letters differ significantly (*p* < 0.01).

**Table 6 animals-14-02152-t006:** Association analysis of haplotypes in RXFP2 with horn traits in Module 2.

Loci	Haplotypes	Numbers	Horn Length (cm)	Horn Base Circumference (cm)	Horn Base Interval (cm)
g.30935807 g.30953070 g.30953972 g.30953997 g.30957852 g.30986879	AACCAAGGCCCC	363	22.17 ± 13.79 ^Aa^	10.04 ± 4.14 ^Aa^	4.07 ± 1.51 ^Aa^
	AACCAAGGCCCT	222	17.49 ± 13.23 ^B^	8.68 ± 4.95 ^B^	3.73 ± 1.95 ^Ab^
	AACCAAGGCCTT	25	12.18 ± 15.80 ^B^	5.50 ± 6.68 ^B^	1.96 ± 2.34 ^B^
	AACCAAGGCTCC	11	10.68 ± 8.21 ^B^	6.91 ± 4.89 ^Ab^	3.41 ± 2.22 ^Aa^
	AACCAAGGCTCT	2	0.00 ± 0.00 ^B^	0.00 ± 0.00 ^B^	0.00 ± 0.00 ^B^
	AACTGAGACCCC	3	3.33 ± 5.77 ^B^	2.33 ± 4.04 ^B^	2.00 ± 3.46 ^Aa^
	AACTGAGACTCC	70	23.78 ± 16.71 ^Aa^	10.16 ± 5.28 ^Aa^	3.86 ± 1.70 ^Aa^
	AACTGAGACTCT	177	8.19 ± 11.83 ^B^	4.06 ± 5.42 ^B^	2.43 ± 2.38 ^B^
	AACTGAGACTTT	48	3.57 ± 8.35 ^B^	2.02 ± 4.40 ^B^	1.38 ± 2.09 ^B^
	AATTGAAATTCT	6	0.33 ± 0.82 ^B^	0.42 ± 1.02 ^B^	2.17 ± 2.40 ^Aa^
	AATTGAAATTTT	18	0.00 ± 0.00 ^B^	0.00 ± 0.00 ^B^	1.11 ± 1.84 ^B^
	AATTGGAATTCT	5	12.60 ± 12.80 ^Aa^	6.70 ± 6.32 ^Aa^	2.40 ± 2.27 ^Aa^
	AATTGGAATTTT	3	0.00 ± 0.00 ^B^	0.00 ± 0.00 ^B^	0.00 ± 0.00 ^B^
	GACCAAGGCCCC	6	16.33 ± 13.93 ^Aa^	8.08 ± 6.48 ^Aa^	2.83 ± 2.32 ^Aa^
	GACTGAGACTCT	66	3.85 ± 8.63 ^B^	2.07 ± 4.64 ^B^	1.07 ± 1.95 ^B^
	GACTGAGACTTT	47	1.48 ± 3.29 ^B^	1.20 ± 2.70 ^B^	0.90 ± 2.07 ^B^
	GACTGAGATTCT	4	0.00 ± 0.00 ^B^	0.00 ± 0.00 ^B^	2.00 ± 2.31 ^Ab^
	GATTGAAATTTT	16	1.63 ± 3.34 ^B^	0.69 ± 2.75 ^B^	0.50 ± 1.37 ^B^
	GATTGGAATTCT	2	0.00 ± 0.00 ^B^	0.00 ± 0.00 ^B^	0.00 ± 0.00 ^B^
	GATTGGAATTTT	15	0.00 ± 0.00 ^B^	0.00 ± 0.00 ^B^	0.00 ± 0.00 ^B^
	GGCTGAGACTCT	2	3.50 ± 4.95 ^Ab^	4.00 ± 5.66 ^Aa^	2.00 ± 2.83 ^Aa^
	GGTTGGAATTTT	4	3.50 ± 4.73 ^B^	2.50 ± 5.00 ^B^	1.25 ± 2.50 ^B^

Within the same line, the mean values of different superscript lowercase letters were significantly different (*p* < 0.05). Within the same line, the mean values of different superscript uppercase letters differ significantly (*p* < 0.01).

## Data Availability

The data utilized in this study are accessible within this article as well as the Appendix A.

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
