# Peer review of "Genome-Wide Association Analyses and Population Verification Highlight the Potential Genetic Basis of Horned Morphology during Polled Selection in Tibetan Sheep"

_animals, 2024, doi:10.3390/ani14152152_

Round 1

Reviewer 1 Report

Comments and Suggestions for Authors

My suggestions for manuscript changes are listed in the .pdf file. The manuscript has the potential and thought the topic is actual, it needs to be refined. The changes are not too extensive, the most important thing is to use a database for GO analysis that has species Ovis aries, improve the Results and the Discussion (include tables that are not commented on and clean the text of unclear sentences), and improve the Figures.

Kind regards,

Reviewer

Comments on the Quality of English Language

Author Response

Comments 1: L24 Provide clarification for G2 (can be in parentheses because it is not possible to know from the summary that it is about offspring).

Response 1: Thank you for pointing this out. We agree with this comment. G2 is the offspring of the polled parent, which we have clarified in line 26 of the manuscript. Where we have modified was highlighted in the revised paper.

Comments 2: L28 Change “horntype” u horn type.

Response 2: Thank you for pointing this out. Agree. We have, accordingly, revised, with the changes highlighted in line 30.

Comments 3: L34 „providing valuable guidance for practical production“ – what would be the guidelines, explain them in few sentences (shortly and clearly) in Introduction and Discussion part.

Response 3: Thanks for reviewer’s suggestion. Agree. We have given a brief explanation in lines 70-73 of the Introduction to the manuscript and in lines 441-444 of the Discussion part. Where we have modified was highlighted in the revised paper.

Comments 4: L70-87 The entire paragraph when author introduce the reader to the issue should be concise and clear. It should be clarified which genotypes are related to the appearance of horns, which to polledness, (sentence 70-71 is unclear). The authors then list the locus for horns at (Merino x Romney) x Merino resource, then list insertions, then new mutations, which have no link to above text, and which, with a lot of stylistic errors (spaces, a comma instead of a period) and the whole paragraph is less meaningful. This part must be changed and rewritten. Below are some of the observed mistakes that need to be changed:

L69 „Breeders have been breeding polledness sheep since the 16th century“ Add reference.

L77 „… sheep[13], One…“ instead comma – dot?!

L77 “One study found that a1833-bp genomic insertion located in the 3’-UTR region 77 of

RXFP2 was shown to be associated with polledness[14][15].” Delete “One” since you have two references and rephrase.

L82 -87 Sentence is too long and should be divided. The author mentions “ …in these studies

L85” – add references.

L 88-89 clarify “excellent gene pool”. The sentence “… some rams and ewes showed no horns”is without link to previous test. Write and explain in a few sentences the importance of the Tibetan sheep and how this research will help in breeding the population because the author mentioned it in the Abstract.

Response 4: Thanks for reviewer’s suggestion. We agree with this comment.

   In line 78 of the manuscript, the reference has been added.

   Yes, here the full stop is more appropriate, has been modified.

   In lines 85-87 of the manuscript, the sentence has been revised.

   In lines 87-90 of the manuscript, the sentence has been revised.

   In line 92 of the manuscript, references have been cited and the sentence has been revised. The two causative mutations of polledness are identified.

   "excellent gene pool" refers to the numerous advantages possessed by Tibetan sheep, such as their ability to withstand cold temperatures, resistance to rough feeding conditions, and adaptability. To address any potential reader skepticism, we have revised it to emphasize its status as one of the local outstanding germplasm resources. Highlighted in lines 98-99 of the manuscript. We deleted the sentences that were not associated with the previous test.

Where we have modified was highlighted in the revised paper.

Comments 5: “early age stages” – add age.

Response 5: Thank you for the comment. Here are the adult sheep. We have revised.

Comments 6: delete “evolutionary process”.

Response 6: Thank you for pointing this out. We agree with this comment. We have deleted.

Comments 7: State the research objective(s) in a separate paragraph.

Response 7: Thank you for the comment. The research objectives are stated in a separate paragraph within lines 108-112 of the manuscript. Where we have modified was highlighted in the revised paper.

Comments 8: L149 For the enrichment analysis, which species did the authors use for GO since the one they used does not include Ovis aries? Additionally, the authors should use a database that includes Ovis aries, such as KEGG.

Response 8: Thanks for reviewer’s suggestion. We use the amino acid sequence of each gene, and then use the eggnog - mapper (2.1.4) (https://github.com/eggnogdb/eggnog-mapper), With the default parameters than eggNOG DB database (5.0.2) (http://eggnog5.embl.de/download/emapperdb-5.0.2/), so as to get the sheep species go term the results of each gene. We have redesigned the enrichment analysis diagram and compiled the RXFP2 gene enrichment GO/KEGG pathway, which offers a more intuitive representation. Consequently, we have revised the corresponding section in the manuscript accordingly. In the Discussion, lines 380-393, we have a discussion of pathways. Where we have modified was highlighted in the revised paper.

Comments 9: L154 Write full name for the abbreviations (first time in text) for CAG, FDR

Response 9: Thank you for pointing this out. Lines 161 and 172 of the manuscript have been added.

Comments 10: L157 “termsin” change “terms in”.

Response 10: Thank you for pointing this out. Line 173 in the manuscript has been revised.

Comments 11: L163-167 make a sketch of measurements and add it to the text,

Response 11: Thanks for reviewer’s suggestion. We have created a graphical representation of the measurement data and appended it to the manuscript. Additionally, comprehensive details regarding the data can be found in supplementary table S11.

Comments 12: L206-208 Use Justify stile for text, change the begining of the sentence to not start with „It“,

Response 12: Thanks for reviewer’s suggestion. We have revised.

Comments 13: L227 – Why was the Structure analysis done and whay number of clusters was chosen to be 5? For example, why not K6 or K7?

Response 13: Thank you for the comment. Population structure analysis is the evaluation of population structure and kinship to determine the statistical model used and obtain the corresponding matrix. The cross-validation error graph in fig.2B indicates that K=2 is the optimal number of clusters, while our value of K is 9; however, I think it is unnecessary to display it.

Comments 14: L230 “…confirming the validity of the sampling” – based on what are they confirmed (add in the text),

Response 14: Thank you for pointing this out. The bivariables in the Kinship plot are perfectly aligned along a straight line, indicating a strong linear relationship. Moreover, the sample exhibits high repeatability, suggesting excellent data reproducibility. Added in the manuscript on lines 254-256. Where we have modified was highlighted in the revised paper.

Comments 15: L233 and 262 is it P-value or p-value,

Response 15: Thank you for pointing this out. The paragraph on the enrichment path has been revised due to graphic adjustments, resulting in the removal of the P-value from the original manuscript.

Comments 16: L270-272 „It is worth noting that the RXFP2 gene exhibits enrichment in 20 signaling pathways co-enriched by three traits”. What is the difference between this and the sentence above 270-272. It seems confusing or needs to be better explained,

Response 16: Thanks for reviewer’s suggestion. The enrichment pathway diagram was redrawn, the corresponding section in the manuscript was revised, and a discussion of this aspect was also included in the discussion section.Where we have modified was highlighted in the revised paper.

Comments 17: L282 “sheep. Revealing” dot – remove,

Response 17: Thanks for reviewer’s suggestion. We have deleted.

Comments 18: Table 3. Add numbers for AA/GA/and GG genotypes and in 1st row delete Frequency because it is number of genotypes or leave Frequency just express number as percentages,

Response 18: Thank you for pointing this out. In Table 4 (added a Table2), we have added numbers for AA/GA/and GG genotypes and in 1st row delete Frequency.

Comments 19: 296“Horns” change into horns,

Response 19: Thank you for the comment. We have revised.

Comments 20: Tables 4 and 5 are not explained anywhere in the text. They must be clarified in the Results and included in the Discussion.

Response 20: Thank you for the comment.The explanation for Tables 5 and 6 (added a Table2) can be found in lines 337, 339-344 of the manuscript and they are cited in the "Discussion" section at lines 418-429. Where we have modified was highlighted in the revised paper.

Comments 21: L331-334 Not clear the meaning of the sentence – please rewrite.

Response 21: Thank you for pointing this out. In lines 400-405 of the manuscript, we have rewritten it.

Comments 22: L346 “The inheritance of horns in sheep is closely associated with sex, as indicated by Wood's findings”Add a reference.

Response 22: Thank you for pointing this out. In line 395 of the manuscript, we have quoted.

Comments 23: L354 “The most effective approach to enhance your flock is by utilizing superior rams. “– This sentence has no connection with text above. What was the point of this sentence? Also the rest of the text (L357) what was the connection with the text above? Rewrite.

Response 23: Thanks for reviewer’s suggestion. Lines 400-405 of the manuscript have been rewritten.

Comments 24: Manuscript has errors such as space (L44, 52, 55, 73) and a capital letter in the middle of the sentence (L52 (injuries), 54 (dehorning), 56, .). I have included only a couple of examples here, and the author.

Response 24: Thank you for the comment. I checked and revised the problems in the manuscript one by one.

Comments 25: The resolution of the figures is very poor, which affects the clarity of the results:

Fig 1. – put Fig. A-D next to each other, it's too big like this. If the resolution is good, the horns and curs will still be visible.

Fig 4. – It is impossible to read the text, even if the reader magnifies them to 200% or 300%, he still cannot make out what is written. Put these figures in the supplemental material if there is no other solution.

Response 25: Thank you for pointing this out. The two graphs have been modified and subsequently replaced.

Reviewer 2 Report

Comments and Suggestions for Authors

Dear aughtors, I have only one question: According to Lei-Lei Li et al., (2021)  according to different geographic conditions, especially altitudes, Tibetan sheep evolved into different breeds!? Which breed exactly is the object of this research?

Author Response

Comments 1: According to Lei-Lei Li et al., (2021) according to different geographic conditions, especially altitudes, Tibetan sheep evolved into different breeds!? Which breed exactly is the object of this research?

Response 1: Thanks for reviewer’s suggestion. Tibetan sheep is one of the three major coarse wool sheep breeds in China, primarily distributed in the high-altitude cold areas of Qinghai, Tibet, Gansu, Sichuan and their adjacent regions. Research indicates that Tibetan sheep gradually expanded from the northeastern part of the Tibetan Plateau 3,100 years ago with the expansion of the Di-Qiang people. Subsequently, they further spread from southwest to central regions 1,300 years ago. These sheep possess unique genetic resources in Qinghai's cold zone and are considered as dominant genetic resources on the Qinghai-Tibet Plateau. Due to geographical distribution and artificial selection influences, Qinghai Tibetan sheep can be categorized into three ecological types: plateau, valley and Eula; among which plateau Tibetan sheep represents the primary type. Our study focuses specifically on plateau Tibetan sheep.

Reviewer 3 Report

Comments and Suggestions for Authors

Dear Authors, in the paper “Genome-Wide Association Analyses Highlight the Potential Genetic Basis of Horned Differentiation During Polled Selection in Tibetan Sheep and Population Verification”, which is very interesting and the results are important for polled Tibetan sheep selection and breeding. In this study, they used GWAS to analyze 103 sheep with different horn phenotype and 382 G2 individuals to check the variations. However, I have a few comments.

I suggest to modify the title, the “Horned Differentiation” seemed no evidence in the manuscript.

Why the author chose the reference genome ncbi_GCF_002742125.1, should it be GCA_017524585.1 for Tibetan sheep or the now one GCF_016772045.2?

Could you please make a new table like table 2 to show which SNPs in RXFP2 as the polled or horned selection signature in other sheep population from the literature, which is new in Tibetan sheep.

I recommend to updated all the figures of this manuscripts, why the author list figure 1A population structure in this manuscript. All the 103 G2 come from the same population by mating the polled rams and ewes (G1), the results seemed the 103 G2 can be clustered different like the natural breeds. Therefore, the figure1A should not be listed.

Author Response

Comments 1: I suggest to modify the title, the “Horned Differentiation” seemed no evidence in the manuscript.

Response 1: Thank you for pointing this out. We agree with this comment. We modify it as “Genome-Wide Association Analyses Highlight the Potential Genetic Basis of Horned Morphology During Polled Selection in Tibetan Sheep and Population Verification”.

Comments 2: Why the author chose the reference genome ncbi_GCF_002742125.1, should it be GCA_017524585.1 for Tibetan sheep or the now one GCF_016772045.2?

Response 2: Thank you for pointing this out. It is preferable to utilize the Tibetan sheep reference genome; however, considering our consistent utilization of this reference genome alongside other omics sequencing, we will continue employing it for comprehensive analysis purposes.

Comments 3: Could you please make a new table like table 2 to show which SNPs in RXFP2 as the polled or horned selection signature in other sheep population from the literature, which is new in Tibetan sheep.

Response 3: Thanks for reviewer’s suggestion. We have generated a novel Table 2 that is derived from the SNP data within the RXFP2 gene of diverse sheep breeds, as documented in the extant literature. These SNPs are chosen as genetic markers for horn presence or absence, and they differ from the SNPs that have recently been identified in Tibetan sheep.

Comments 4: I recommend to updated all the figures of this manuscripts, why the author list figure 1A population structure in this manuscript. All the 103 G2 come from the same population by mating the polled rams and ewes (G1), the results seemed the 103 G2 can be clustered different like the natural breeds. Therefore, the figure1A should not be listed.

Response 4: Thanks for reviewer’s suggestion. Figures 1 and 4 have been revised and Figure 1 has been new added. Figure 1 depicts the morphology of 103 G2 sheep with different horn types. We agree with this comment. We have deleted. 

Reviewer 4 Report

Comments and Suggestions for Authors

Dear Author(s),

Please provide the revisions as given below.

Best Regards.

Simple summary section:

The simple summary is sufficient.

Abstract section:

Abstract is clear and understandable.

Introduction section:

Line 46-49: Please add a citation for this expression.

Line 57, Line 71, Line 78: Please re-check the citation order according to the journal rules.

After Line 66: Please mention about GWAS in the introduction section.

Material and methods section:

This section is successful. But the section needs some revisions.

Line 153: Please move the equation to the new line, not in the text.

Line 194: Please explain why you used this general linear model, and in addition, please use the subscript of each feature in the equation.

Results section:

Line 243: Please explain the Fig 2 (A, B, C, D). I think the Fig 2.D is not successfully for explaining the variance with 3 PCs. I think these should be explained.

Line 300, Line 320 and 321: I think the titles of Table 3, Table 4 and Table 5 are wrong. Because correlation analysis results are never over ±1. Please check again.

Discussion section:

Discussion is enough.

Author Response

Comments 1: Line 46-49: Please add a citation for this expression.

Response 1: Thank you for pointing this out. On line 48 of the manuscript, we have added.

Comments 2: Line 57, Line 71, Line 78: Please re-check the citation order according to the journal rules.

Response 2: Thank you for pointing this out. The citation order has been re-adjusted.

Comments 3: After Line 66: Please mention about GWAS in the introduction section.

Response 3: Thanks for reviewer’s suggestion. WGS is mentioned in lines 70-73, 93-96, and 108 of the manuscript's introduction.

Comments 4: Line 153: Please move the equation to the new line, not in the text.

Response 4: Thanks for reviewer’s suggestion. We have moved the equation to the next line.

Comments 5: Line 194: Please explain why you used this general linear model, and in addition, please use the subscript of each feature in the equation.

Response 5: Thank you for the comment. The general linear model is capable of assessing the interplay between variables as well as the independent influence of individual factors. It affords comprehensive regression analysis and analysis of variance (ANOVA) for a array of dependent variables, contingent upon one or more factor variables or covariates. The correlation analysis between genes and traits is predicted by the general line model, such as the literature: Jiang, R.; Cheng, J.; Cao, X.-K.; Ma, Y.-L.; Chaogetu, B.; Huang, Y.-Z.; Lan, X.-Y.; Lei, C.-Z.; Hu, L.-Y.; Chen, H. Copy Number Variation of the SHE Gene in Sheep and Its Association with Economic Traits. Animals 2019, 9, 531. https://doi.org/10.3390/ani9080531. Subscripts have been used for each feature in the equation.

Comments 6: Line 243: Please explain the Fig 2 (A, B, C, D). I think the Fig 2.D is not successfully for explaining the variance with 3 PCs. I think these should be explained.

Response 6: Thank you for pointing this out. Population structure analysis is the evaluation of population structure and kinship to determine the statistical model used and obtain the corresponding matrix. The cross-validation error graph in fig.2B indicates that K=2 is the optimal number of clusters. PCA constitutes a linear dimensionality reduction method that transcends high-dimensional datasets into a reduced-dimensionality space, thereby preserving the data's intrinsic variability. Although the distributional characteristics of the data may exhibit alterations in the lower-dimensionality realm, such discrepancies are merely indicative of the data's inherent structure rather than being a consequence of any categorical differentiation. We have also provided additional clarifications in lines 254-256 of the manuscript. Where we have modified was highlighted in the revised paper.

Comments 7: Line 300, Line 320 and 321: I think the titles of Table 3, Table 4 and Table 5 are wrong. Because correlation analysis results are never over ±1. Please check again.

Response 7: Thank you for pointing this out. In 330, 353, and 354 lines of the manuscript, the title has been amended to reflect a more accurate term, 'correlation analysis,' rather than the previously stated 'correlation analysis. Where we have modified was highlighted in the revised paper.

Round 2

Reviewer 1 Report

Comments and Suggestions for Authors

Dear Authors, dear Editor,

The manuscript has been significantly revised and improved. As a reviewer, I have no new comments. Perhaps just one suggestion, which is entirely up to the author: the first version included Figure of horns that are not included in the new version; I can suggest adding Figure to the xls table as Supplement material if it does not take a lot of time. But even without that, the work is correctly corrected and can be published in this form.

Kind regards,

Reviewer

Reviewer 4 Report

Comments and Suggestions for Authors

Dear Author(s),

Corrections were provided by the authors.

Best Regards